# MAD-Sherlock: Multi-Agent Debates for Out-of-Context Misinformation Detection

## Abstract

One of the most challenging forms of misinformation involves the out-of-context (OOC) use of images paired with misleading text, creating false narratives. Existing AI-driven detection systems lack explainability and require expensive finetuning. We address these issues with MAD-Sherlock: a **M**ulti-**A**gent **D**ebate system for OOC Misinformation Detection. MAD-Sherlock introduces a novel multi-agent debate framework where multimodal agents collaborate to assess contextual consistency and request external information to enhance cross-context reasoning and decision-making. Our framework enables explainable detection with state-of-the-art accuracy even without domain-specific fine-tuning. Extensive ablation studies confirm that external retrieval significantly improves detection accuracy, and user studies demonstrate that MAD-Sherlock boosts performance for both experts and non-experts. These results position MAD-Sherlock as a powerful tool for autonomous and citizen intelligence applications.

## 1 Introduction

Our growing dependence on online channels for news and social networking has been complemented by a surge in exploits of digital misinformation (Aslett et al., 2024; Hasher et al., 1977; Brashier & Marsh, 2020). While many manipulation techniques pose serious threats, one of the most prevalent methods for creating fake online content is the out-of-context (OOC) use of images (pbs). This involves using unaltered images in a misleading, false context to convey deceptive information, a strategy that requires minimal technical expertise. Indeed, the problem of OOC misinformation detection requires a complex understanding of the relationship between the text and image and the ability to identify when they do not go together. Identifying these minute inconsistencies is a time-consuming and high-effort task for humans. A study by Sultan et al. (2022) shows that time pressure reduces the ability of human beings to detect misinformation effectively, further adding to the scalability issues in human expert detection.

Therefore, attention has turned to AI-driven tools that can help human experts recognise instances of OOC image-based misinformation at scale. Unfortunately, conventional deep learning forensic techniques (Castillo Camacho & Wang, 2021; Heidari et al., 2024; Zhu et al., 2018; Amerini et al., 2021; Hina et al., 2021), which target detecting manipulations such as PhotoShop editing (Tolosana et al., 2020; Masood et al., 2023; Farid, 2016; Wang et al., 2019) and AI-generated (or manipulated) fake images called Deepfakes (mit), rely on spotting artifacts from image or text tampering. In contrast, OOC detection demands cross-contextual reasoning, as the deception arises from the misalignment between the legitimate image and its falsely associated textual content.

Pretrained Large Multimodal Models (Liu et al., 2024b; OpenAI & et al., 2024; Li et al., 2019; Radford et al., 2021, LMMs) provide a promising direction for detecting OOC use of images for their ability to process both text and image content in tandem. However, using LMMs directly for OOC detection presents several challenges, particularly in the news domain. For instance, news articles often include images that are not directly related to the article's content. An article about the 2024 U.S. presidential candidates, for example, might feature a close-up of Donald Trump from an unrelated online database. Although the image was taken outside the election period, it is not considered OOC since it doesn't misrepresent the article's context. Such cases complicate LMMs' ability to accurately identify OOC usage based solely on their pre-trained knowledge, as this knowledge may be outdated or insufficiently detailed.

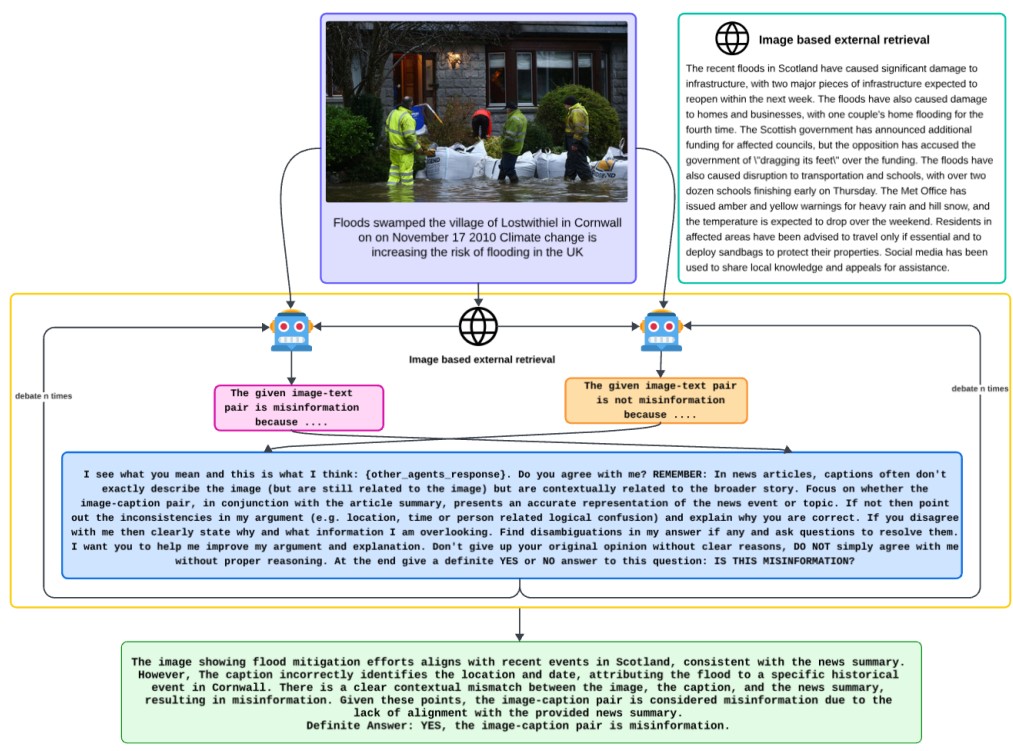

Figure 1: **Overview of MAD-Sherlock:** Two or more independent agents see the same image-text input and are tasked with detecting whether the input is misinformation or not. After the agents form their independent opinions, they participate in a debate until they converge on the same response or when $n$ debate rounds are completed (whichever is earlier).

Moreover, even with recent advancements, LMMs are capable of hallucinating and, hence, generating false information (Bai et al., 2024; Liu et al., 2024a). While rapidly improving, they sometimes fail to understand user instructions and intent correctly. We show that off-the-shelf LMMs indeed suffer from these issues, reducing their ability to detect OOC misinformation in practice. While prior work (Qi et al., 2024) has shown that off-the-shelf models can be improved using task-specific fine-tuning, this approach is resource-intensive and requires continual updating to keep up with recent events. Moreover, detecting OOC images only solves part of the problem. The real value lies in being able to *explain* the OOC use of pictures in human-readable form. It can be instrumental for human validators to observe the model's line of logic and gain better insight into, and trust in, the classification process.

In this work, we propose a novel LMM-based post-training approach for scalable OOC misinformation detection that simultaneously improves contextual reasoning, provides in-built explainability, and achieves state-of-the-art detection accuracy even without task-specific fine-tuning (see Section 3). Specifically, our framework *MAD-Sherlock: a **M**ulti-**A**gent **D**ebate system for OOC Misinformation Detection* frames the detection problem as a dialectic debate between multiple LMM agents, where, in contrast to prior work (Minsky, 1988; Li et al., 2023a; Du et al., 2023a; Khan et al., 2024)), agents have access to external information retrieval.

Compared to single-agent chain-of-thought approaches (Wei et al., 2024), the use of multiple agents allows for a clean separation of agent contexts, decentralisation of action spaces, and opportunities for parallel computation (Schroeder de Witt et al., 2020; Du et al., 2023b). In addition, due to its compositional nature, both additional human and autonomous agents can be dynamically added to the multi-agent reasoning process, allowing the use of MAD-Sherlock as an interactive tool for human experts. To the best of our knowledge, no prior work has used debating LMMs for detecting and *explaining* OOC image use.

We perform a comprehensive empirical evaluation of our method (see Section 4), including the study of multiple debate configurations. To optimise OpenAI API use, we utilize an experimental pipeline where preliminary experiments are performed using the open-source LLaVA model (Liu et al., 2024b), which is only later replaced with GPT-4o (OpenAI) to achieve state-of-the-art performance. We find that *MAD-Sherlock* outperforms both prior work and novel baselines that we introduce, is more robust to various failure modes, and produces coherent explanations that help both human experts and non-experts significantly improve their detection accuracy in user studies. We identify both access to external information retrieval and complete freedom of opinion as key ingredients to MAD-Sherlock's performance. Finally, we discuss current limitations of our method and propose future work toward overcoming the scalability challenges in large-scale online OOC misinformation detection.

## 2 RELATED WORK

Recent work has focused on using joint image-text representations to classify an instance as OOC. Aneja et al. (2022) follow a self-supervised approach to assess whether two captions accompanying an image are contextually similar. They enforce image-text matching during training by formulating a scoring function to align objects in the image with the caption. During inference, they use the semantic similarity between the two captions to classify them as OOC or not. The increased reliance on textual content limits the capabilities of this approach. This work also does not provide explanations for model predictions and is, therefore not interpretable. Moreover, this method works for image caption pairs where captions have information about objects in the image. This is not always the case with news articles (our domain of application), where captions can often just be related to the main content of an article rather than precisely describing the objects in the image. Appendix A.2 shows an example of the same.

Abdelnabi et al. (2022) present the Consistency Checking Network (CCN) in which they emulate different aspects of human reasoning across modalities for misinformation detection. This method uses evidence related to the image-text pair aggregated from the Internet. The CCN consists of memory networks to assess the consistency of the image-caption pair against the retrieved evidence and a CLIP (Radford et al. (2021)) component to evaluate the consistency between the image and caption pair. The use of external evidence to better inform model decisions is an important idea and also explains the superior classification performance of CCN when compared to other methods. This method also lacks the explainability component.

Zhang et al. (2024) extend the neural symbolic method (Yi et al. (2019); Zhu et al. (2022)) to propose an interpretable cross-modal misinformation detection model to provide supporting evidence for the output prediction. They use symbolic graphs based on the Abstract Meaning Representation (Banarescu et al. (2013)) of textual and visual information to detect OOC image use. Zhou et al. (2020) introduce Similarity Aware Fake news detection (SAFE), where neural networks are used to learn features of text and visual news representations. Their representations and relationships are jointly learned and used to predict fake news. Wang et al. (2018) introduce EANN: Event Adversarial Neural Networks to derive event invariant features which can be used to detect fake news that has recently been generated. EANN uses adversarial training to learn multi-modal features independent of news events. These methods require pretraining from scratch and, therefore, don't benefit from the advanced reasoning capabilities and world knowledge of large pretrained models.

Shalabi et al. (2023) use synthetic multi-modal data to establish the authenticity of image-text pairs. They use BLIP-2 (Li et al. (2023b)) to generate a caption for the original image and Stable Diffusion (Rombach et al. (2022)) to generate an image for the given original caption. This synthetic data is then used to reason that if the original image and caption are OOC, then the original and generated images should also be OOC as well as the original and generated text. This method relies on synthetic multi-modal data generation, which not only adds an additional computational overhead but also increases dependence on often unreliable synthetically generated data. Therefore, this method can suffer from issues related to generation models, including potential biases that these models may possess. This method also lacks interpretability.

Sniffer (Qi et al. (2024)) is the closest to our work. It uses the InstructBLIP (Dai et al. (2023)) model to detect OOC image use and provide an explanation for its prediction. It makes use of internal and external knowledge using entity extraction APIs and image-based web searches. Information from

all the sources is given to an LLM to predict and explain if an image has been used OOC. Sniffer only uses basic textual information, such as news article titles from websites, to form its external knowledge base. It also requires extensive training to adapt the model to the news domain which adds additional computational overhead and also restricts the generalization abilities of the model to other domains.

# 3 METHODOLOGY

We present an explainable misinformation detection system, MAD-Sherlock, which jointly predicts and explains instances of misinformation. Figure 1 illustrates our approach. To the best of our knowledge, all prior work except Qi et al. (2024) provide predictions without explanations, and no prior work uses multiple models to approach this problem. We present a novel methodology that involves multiple multi-modal models debating against each other in order to decide if an image-text pair is misinformation or not. In this work, we aim to answer the question:

*Can debating multi-modal models, when equipped with external context, be used to solve the problem of explainable misinformation detection by picking up on minute contextual inconsistencies?*

We use detailed external information to inform the model's predictions through the external retrieval module, which utilises reverse image-based search in order to provide the agents with external real-world context related to the image-text pair. We carry out our experiments with the GPT-4o (OpenAI) model to achieve state-of-the-art performance on the misinformation detection task while also providing detailed and coherent explanations for the predictions. We achieve this without any domain-specific fine-tuning, thus ensuring easier and faster generalization to other domains in addition to low computational overhead.

## 3.1 DEBATE MODELLING

Analogous to real-world conversations, communication between two AI agents can also be structured in a myriad of ways. We explore multiple debating strategies to structure the conversation between agents, all of which are tested and evaluated in our experiments. Instead of simple back-and-forth conversations, we opt for a debating set-up in which agents are asked to frame their own opinions and then defend them to other agent(s). We observe this facilitates more involved and detailed discussions among the models.

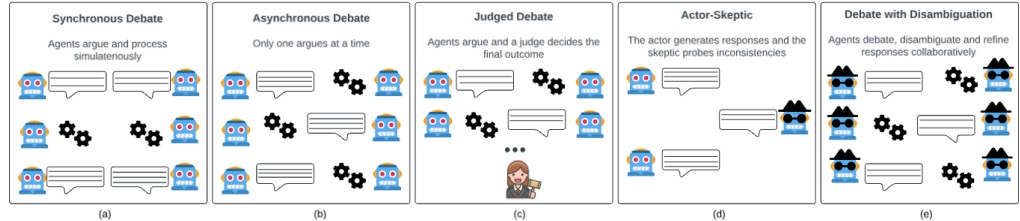

Figure 2: **Debating Strategies:** We experiment with multiple debating strategies. The asynchronous debate setup where agents argue one after the other and take turns presenting their arguments is the best configuration.

**Asynchronous Debate (not) against Human:** This is one of the core setups we test in our experiments. We define an asynchronous debating strategy in which models wait for the other participants' responses before generating their own. Figure 2 (a) and (b) show synchronous and asynchronous debating structures, respectively. While synchronous debates, where all participants speak at once, can be faster and computationally more efficient, we opt for an asynchronous setting where each model response is based on previous responses of other models. We observe that this method of structuring the debate works better since models are able to pick up on contextual ambiguities in their responses in a more organised and structured way which is crucial to the process of misinformation detection.

An important point to note for this setup is the way we structure model prompts. The debating models are not aware that they are debating other AI agents. The prompts are structured such that each debating model believes it is talking to a human.

**Judged Debate:** We also experiment with an asynchronous debate setup with a judge. Figure 2 (c) shows this setup. In this setup, models participate in an asynchronous debate as usual however, the final decision is made by a judge at the end of the debate. Models are incentivised to structure their arguments in a way that makes them most convincing to the judge. We structure this debate configuration similar to Khan et al. (2024), where the judge does not have access to all the external information and has to rely only on the debate transcript to decide the final answer.

**Actor-Skeptic:** In this setup, only one agent, the *actor*, is tasked with deciding whether a given image-text pair is misinformation. The agent generates a response which a *skeptic* then evaluates. The skeptic is tasked with finding logical errors in the actor's argument and asking follow-up questions to disambiguate the actor's response. It is important to note that neither the skeptic nor the actor has access to the ground truth. This setup does not benefit from an ensemble since both models assume different roles and only one agent is tasked with generating the final answer.

**Debate with Disambiguation:** Improving on the actor-skeptic method, in this setup, we allow all agents to act as actors *and* skeptics. Models are tasked with not only generating their own responses but also disambiguation queries to refine further or refute the other agents' responses. These disambiguation queries are then used to search the Internet to obtain information to refine model outputs further. We are the first to propose this debate setup and while it does not achieve the best results in this work, we believe future research can greatly benefit by refining this setup further.

Through empirical testing of all the described debate set ups, we identify asynchronous debate— where the model believes it is debating against a human rather than an AI agent—as the most effective configuration.

## 3.2 PROMPT ENGINEERING

The debate structure is substantiated through prompt engineering. Figure 1 shows that the first stage of our method requires for each AI agent to generate an independent response to whether the given image-text pair is misinformation. Each agent must take into account the external context related to the image obtained through the external information retrieval module. Specifications of the various prompts used in this work can be found in Appendix A.3. An initial prompt provides the agent with a summary of the news articles related to the image and, based on it, asks the agent to classify the image-text pair as misinformation or not. The prompt asks the agent to focus on certain details in the image, such as watermarks, flags, etc. We observe that images used in news articles often contain minute yet crucial details which can be used to inform the final decision about whether the image actually belongs to the news articles. Therefore, prompting the agent to pay special attention to these details further helps detect inconsistencies.

Once the agents have formed independent opinions about the image-text pair, they must then participate in a debate. While the same prompt can be used for each debate round, we provide a different prompt for the first round to allow each agent to understand the changing nature of the conversation. Responses from other agents are provided as a part of the prompt, and the agent is asked to agree or disagree. The agent must also clearly state the reasoning process behind its argument. The prompt further requires the agent to identify ambiguities in the other agents' reasoning. This allows the agent to closely analyse the responses from other agents and use them to inform its own decision.

A separate prompt is then used to facilitate all rounds of the debate following round one. It takes into account the other agent's suggestions from the previous round and, after refining its own response, asks the agent to point out any inconsistencies in this new response. We note that in such a scenario, agents are prone to simply agreeing with each other and repeating the other agent's response. This is especially the case when agents believe they are conversing with a human. However, this tendency of the agents to easily give up their own opinions and simply agree with the other participants does not facilitate an advantageous debate, and the agents do not discover any new information related to the image-text pair. Therefore, the debate prompt also contains clear and explicit instructions asking

the agent not to simply agree with the presented response unless it has an acceptable reason to do so. We find this helps the agents develop stronger stances and reluctance to blind agreement.

## 3.3 EXTERNAL INFORMATION RETRIEVAL

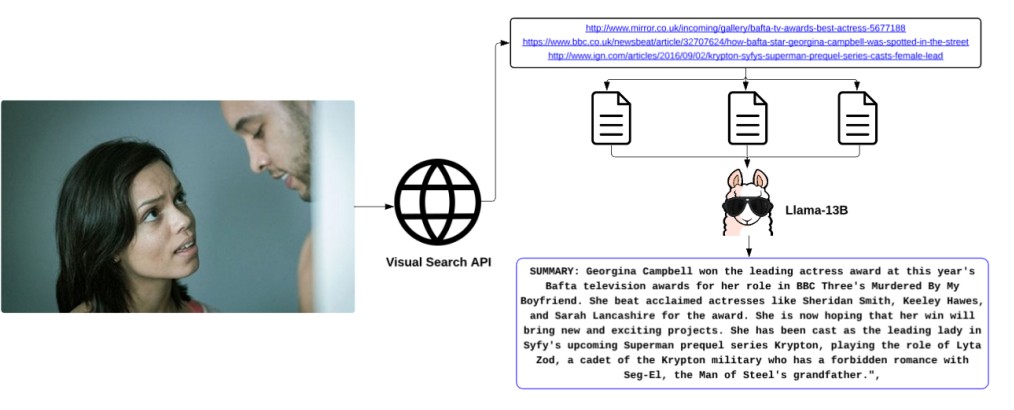

Figure 3: **Structure of the external information retrieval module**: We use the Bing Visual Search API (vis) to obtain web pages related to a given image, which are then summarised using Llama-13B (Touvron et al. (2023)). This summary is then passed to the debating agents as a part of the initial prompt.

Since a model's world knowledge is limited to its training data (and hence a particular time frame), incorporating external retrieval allows the model to access information beyond this training data (and time frame). Previous work makes use of pre-existing external retrieval-based datasets (Abdelnabi et al. (2022)) to supplement external information related to an image-caption pair. However, we find this information lacking in detail since it is limited to the title of a news article. Agents can greatly benefit from the knowledge of the entire news article and its content rather than just the title when making a decision about whether a given image-caption pair, when considered in the context of the news article, is misinformation. To this end, we propose our own external information retrieval module. We observe a significant improvement in accuracy after incorporation of the external information retrieval module into the pipeline. The module is implemented in two stages:

### 3.3.1 API-BASED INFORMATION RETRIEVAL

The Bing Visual Search API (vis) is used for the task of obtaining web pages related to a given image. A given image from the dataset is used to obtain a list of web pages completely and partially related to the image. We take the top three matching web pages in which the image appears. We believe these web pages contain sufficient information to allow the agent to develop a general understanding of the context in which the image is originally used. Since the community-accepted dataset for this task; NewsCLIPpings (Luo et al. (2021)), contains images from articles published more than ten years ago, some images do not result in any web pages or viable search results. In such a scenario, we simply do not pass any external context to the agent and only rely on the agent's existing knowledge base. Since this is not the case for a significant fraction of examples in the dataset, it does not adversely affect system performance.

### 3.3.2 SUMMARIZATION USING LLM

Once the top three web pages have been identified, we scrape the text from the web pages to obtain the textual information related to the context in which the image appears on the Internet. The compiled textual information is often too long to be passed directly to the agent, and hence, we use the Llama-13B (Touvron et al. (2023)) language model to summarize this information. The summaries obtained from the LLM only focus on the most important parts of the text and hence also allow agents to develop a more focused understanding of the external context. While this method of summarization works for most samples in our dataset, there are some examples where the obtained web

pages are not in the English language, and the LLM struggles with summarization. In this regard, we add an additional check that ignores text from web pages in languages other than English. While this restricts our system to the English language, it does not adversely affect system performance due to the distribution of the dataset, which consists of images mostly taken from English-language news articles. Multi-lingual support can be achieved by first translating text to English and then summarizing it.

### 3.4 COHERENT REASONING

All the different components of MAD-Sherlock are brought together in this stage of the pipeline. Each multi-modal agent is employed to participate in the best-debating set-up with the relevant prompts and is asked to detect a given image-text pair as misinformation and provide an explanation for the same. The agents also have access to external information related to the image through the external retrieval module. The final decision of the system is obtained once the debate terminates, which is after a certain number of debate rounds or after all agents converge to a common response, whichever is earlier.

## 4 EXPERIMENTS AND RESULTS

### 4.1 DATASET

We perform a series of experiments and report results on the NewsCLIPpings dataset (Luo et al. (2021)). The dataset is built based on the VisualNews (Liu et al. (2020)) dataset, which consists of image-caption pairs from four news agencies: BBC, USA Today, The Guardian and The Washington Post. The NewsCLIPpings dataset is created by generating OOC samples by replacing an image in one image-caption pair with a semantically related image from a different image-caption pair. CLIP (Radford et al. (2021)) is used to retrieve semantically similar images for a given caption. We report results on the Merged-Balanced version of the dataset, which has balanced proportions of all the retrieval strategies and positive/negative samples. The training, validation and test sets have 71,072, 7,024 and 7,264 samples, respectively.

### 4.2 EXPERIMENTAL SETUP

All experiments were run on 8 A40 (46GB) Nvidia GPU server. The estimated cost of processing one data sample using MAD-Sherlock is \$0.24 and it takes between 5 to 15 seconds to do so.

**Debate Setup:** We conduct experiments to select the best debating configuration using the LLaVA model (Liu et al. (2024b)). The experiments are carried out on a smaller subset containing 1000 test samples of the main NewsCLIPpings test dataset. All experiments are run for $k = 3$ rounds or until the agents converge (whichever is earlier).

**External Retrieval Module:** We use the Bing Visual Search API (vis) to run an image-based reverse search. Using the API we select the top $k = 3$ pages in which the image appears and scrape the text from them using the Newspaper3k library (new). Finally, we use Llama-13B (Touvron et al. (2023)) to summarise the text obtained from the top $k = 3$ web pages. This step is crucial since the web pages are usually news articles which contain large amounts of text which, when scraped and passed directly to the model, can exceed its maximum token length.

**Baselines and Prior Work:** We compare MAD-Sherlock to existing pretrained multi-modal baselines including CLIP (Radford et al. (2021)), VisualBERT (Li et al. (2019)), InstructBLIP (Dai et al. (2023)) and LLaVA (Liu et al. (2024b)). We also compare performance against GPT-4o (OpenAI & et al. (2024); OpenAI). The models are presented with the image and caption pair and asked if the pair is misinformation. The models are further prompted to explain their reasoning. We also show results for two baseline methods trained from scratch, namely EANN (Wang et al. (2018)) and SAFE (Zhou et al. (2020)). We further compare MAD-Sherlock to DT-Transformer (Papadopoulos et al. (2023)), CCN (Abdelnabi et al. (2022)), Sniffer (Qi et al. (2024)), VINVL (Huang et al. (2024)), SSDL (Mu et al. (2023)) and Neuro-Sym (Zhu et al. (2022)).

### 4.3 RESULTS

We present results for the experiments conducted to select the best debate setup as well compare the performance of MAD-Sherlock against existing methods. We use classification accuracy as the primary performance metric for comparison based on quantitative analysis.

#### 4.3.1 COMPARING DEBATE SETUPS

We compare multiple debating setups using the LLaVA model, to select the best one for comparison with other works and further experimentation.

| Debate Setup | Accuracy | Precision | Recall |
|---|---|---|---|
| Async_Debate$_{AI}$ (believes debating AI) | 75.2 | 54.5 | 86.4 |
| Async_Debate$_{human}$ (w/o external info) | 77.1 | 68.4 | 89.3 |
| **Async_Debate$_{human}$ (w external info)** | **86.2** | **82.6** | **90.6** |
| Actor-Skeptic | 69.5 | 66.1 | 69.4 |
| Judged Debate | 66.7 | 66.7 | 61.5 |
| Debate with Disambiguation | 77.8 | 74.7 | 82.6 |

Table 1: **Performance comparison between different debate setups:** The Async_Debate$_{human}$ where the model has external context and believes it is debating a human being is the best setup.

Table 1 shows that the Async_Debate$_{human}$ setup where the agent has access to external information performs the best of all the debating configurations. We also report results for the Asynchronous Debate setup without access to external information to emphasise the importance of external information for the problem of misinformation detection in the news domain. The external retrieval of information significantly boosts performance. All following debate set-ups, therefore, use external information related to the image-caption pair as a part of their initial prompt. We also observe a significant performance increase when the agent believes it is conversing with a human instead of another AI agent. Qualitatively, the agent considers the other agent's responses more critically and with more seriousness when it believes that the agent is a human. Further, the Asynchronous Debate setup benefits from the ensemble of agents which is not present in the actor-skeptic setup, where only one agent is responsible for generating the responses. The generation of disambiguation queries within the same response, confuses the agents and even deviates them from their own chain of thought. We believe this accounts for the counter-intuitively performance of this method where agents perform worse with more information. The judged debate setup focuses on enforcing agents to structure their responses in a way that will convince a judge. The agents also debate with opposite stances and do not have the option of changing their stance mid-debate. This can further confuse the judge and lead to incorrect decisions. This is resolved in the Async_Debate$_{human}$ set up where agents are given complete freedom over their initial opinions, as well as their opinions during the debate. If they believe they are convinced by the other agents' arguments, they can choose to change their response and the debate ends. Based on the results from Table 1, we choose the best-performing debate set-up, i.e. Async_Debate$_{human}$ (with external information) as the debate configuration for further experimentation and comparison.

#### 4.3.2 PERFORMANCE COMPARISON

We present our results on the NewsCLIPpings dataset against existing out-of-context detection methods discussed in section 4.2.

Table 2 shows the comparison between our system and existing methods. We report state-of-the-art performance when using our proposed debate configuration with the GPT-4o (OpenAI & et al. (2024); OpenAI) model. Sniffer (Qi et al., 2024), being the only work comparable in performance to ours, is finetuned extensively to adapt it to the NewsCLIPpings dataset. While we do not provide a quantitative assessment of explanations by MAD-Sherlock, we do believe our system produces more coherent, detailed and comprehensive explanations when compared to other baselines. This is attributed to the fact that in a multi-agent setup, we have multiple context windows which leads to more coherent and relevant final explanations. We leave the detailed analysis of these explanations and the development of the associated metrics as future work. We also note that the debate paradigm

| Model | Accuracy↑ |
|---|---|
| SAFE | 50.7 |
| EANN | 58.1 |
| VisualBERT | 54.8 |
| CLIP | 62.6 |
| InstructBLIP | 48.6 |
| LLaVA | 57.1 |
| GPT-4o | 70.7 |
| DT-Transformer | 77.1 |
| CCN | 84.7 |
| SSDL | 65.6 |
| VINVL | 65.4 |
| Neuro-Sym | 68.2 |
| GPT-4o[#] (w internet access) | 86.00 |
| Sniffer (w finetuning) | 88.4 |
| Sniffer (w/o finetuning) | 84.5 |
| **MAD-Sherlock (ours)** | **90.17** |

Table 2: **Performance comparison between our model and baselines:** MAD-Sherlock (with GPT-4o) out performs all related work. Note: the GPT-4o[#] setup is identical to MAD-Sherlock with the absence of a multi-agent debate, here only a single agent which has access to external information is considered and the results are reported on a smaller heldout test set of 1000 samples.

in itself is essential to the system performance. We observe a drop in performance and quality of explanations when using an identical system configuration but with a single model.

We also note that single multi-modal models, including VisualBERT, CLIP, InstructBLIP, LLaVA and GPT-4o do not perform at par with other related work. This can be attributed to the necessity for external context for misinformation detection in the news domain and the lack of diverse perspectives that arise naturally in a multi-agent framework. Therefore, these standalone models, while promising, currently are unable to detect misinformation effectively. These models require additional integration into more comprehensive pipelines, as done in this work. In line with previous work, we also note that baselines trained from scratch, such as SAFE (Zhou et al. (2020)) and EANN (Wang et al. (2018)) perform worse than pretrained multi-modal models. This further concretizes the fact that image-based OOC detection in the news domain requires strong world knowledge as well as advanced multi-modal reasoning capabilities.

## 5 USER STUDY

We conducted a user study to evaluate the effectiveness of our system in detecting and explaining misinformation. While it is easy to quantify model performance in terms of misinformation detection, there are no effective metrics to assess the quality of the explanations generated by the model. Therefore, in order to perform a thorough analysis of the system performance, a user study is essential. For a deeper analysis we further grouped the participants based on their profession into three groups, namely: Journalists, AI Academics (studying AI) and Others. Further details regarding the study setup and participant groups can be found in Appendix A.4.

In the study, participants were shown ten image-text pairs and were asked to decide if the image and caption, when considered together, were misinformation or not. They were also asked to provide a confidence rating for their answer on a scale of 0-10, with 10 being the highest confidence level. For each image-text pair, after the participants provided their initial answers, they were shown AI insights about the same image-text pair. These AI insights were the final output explanations from MAD-Sherlock. Participants were then asked to reconsider their answers in light of the new information from the AI agent. Table 3 shows that average system performance is better than the average human performance for both cases where the participants have access to AI insights and where they do not. Therefore, MAD-Sherlock can be used as a reliable assistive tool for OSINT research for detecting and explaining misinformation with little or no human intervention.

| Study Setup | Average Accuracy↑ |
| --- | --- |
| Humans | $60.3 \pm 13.5$ |
| Humans+MAD-Sherlock | $76.7 \pm 12.2$ |
| **MAD-Sherlock** | **$80.0 \pm 0.0$** |

Table 3: **Performance comparison between different study setups:** MAD-Sherlock outperforms humans with and without AI assistance.

We further observe that the average human accuracy for the misinformation detection task increases by more than 27% concretizing the fact that AI insights from our model do actually improve human efficiency in detecting misinformation. We also observe interesting patterns for group-wise analysis which we believe would be valuable for future work. Table 4 shows that the performance of all groups improves significantly and is not far off from that of professional journalists. The average confidence level (out of 10) is comparable across all the groups before and after considering MAD-Sherlock insights and generally increases. Therefore, we conclude that MAD-Sherlock can significantly uplift non-expert performance and hence can be useful in citizen intelligence applications.

| Group | Avg acc ↑ (only human) | Avg conf ↑ (only human) | Avg acc ↑ (with MAD-Sherlock) | Avg conf ↑ (with MAD-Sherlock) |
| --- | --- | --- | --- | --- |
| Journalists | $70.0 \pm 1.4$ | $4.3 \pm 2.1$ | $82.2 \pm 0.9$ | $5.3 \pm 1.3$ |
| AI Academics | $60.7 \pm 1.4$ | $3.2 \pm 0.8$ | $79.3 \pm 1.3$ | $5.8 \pm 1.4$ |
| Others | $56.7 \pm 1.5$ | $3.9 \pm 1.2$ | $71.7 \pm 1.1$ | $5.8 \pm 1.4$ |

Table 4: **Performance comparison between different participant groups:** All groups show performance improvement with MAD-Sherlock. AI Academics are able to perform nearly at par with professional journalists after considering insights from MAD-Sherlock.

# 6 CONCLUSION AND FUTURE WORK

Misinformation detection has become a pressing issue in recent times. With the ever-advancing capabilities of vision and language models, the detection of OOC image use has become a very difficult task. In this work, we explore the question of whether it is possible for multiple AI agents to pool their contextual knowledge and converge to a common prediction in order to identify instances of misinformation. We identify $\texttt{Asynchronous\_Debate}_{\text{human}}$ as the most optimal communication setup for AI models. We observe significant performance improvement when the models believe they are debating against a human instead of another AI agent. We observe that in this setup, models tend to be more involved and open to changing their opinions. Our method also allows for agents to have freedom of opinion which they may change mid-debate. Agents in such a setting show enhanced abilities to critically evaluate an argument and pick up on minute inconsistencies.

Our final system, MAD-Sherlock, achieves state-of-the-art performance on the misinformation detection task. Further, owing to our advanced external retrieval module, MAD-Sherlock provides clear, coherent and detailed explanations. As a result, MAD-Sherlock significantly improves the OOC misinformation detection performance of both human experts, and non-experts.

We identify several promising avenues for future research in this field. The research community would benefit from a continuously updated benchmark dataset, incorporating more recent news articles and subtler inconsistencies. A direct extension of this work involves applying our methods to video-text pairs and supporting multi-lingual content. Future extensions of this work could further validate our findings by leveraging more advanced and refined models in the summarization pipeline which we believe would further improve system performance. It is also worth comparing MAD-Sherlock to systems using multi-agent collaboration with external information retrieval.

Finally, while we conducted extensive user studies with MAD-Sherlock, deploying it on a larger scale in professional environments and within the citizen intelligence community will provide valuable insights into its real-world performance, uncovering new opportunities for improvement. For an analysis of limitations, please refer to Appendix A.1.

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

## A  APPENDIX

### A.1  LIMITATIONS

Despite the strong performance of MAD-Sherlock, several limitations remain. First, while our model excels at detecting out-of-context image-text pairs, its reliance on external retrieval can lead to reduced accuracy when relevant context is unavailable or difficult to retrieve. Second, the quality of explanations is constrained to textual outputs, limiting multi-modal explanation capabilities such as image or video integration. Third, the system's performance is sensitive to hyperparameter tuning, including the number of debate rounds and agents, which may require further optimization for broader use cases.

Additionally, while our user studies provided valuable insights, large-scale deployment in diverse, real-world settings, such as professional or citizen intelligence environments, is necessary to fully assess the method's robustness and scalability. Finally, our dataset, though comprehensive, primarily focuses on English-language news, limiting the generalizability of the system across non-English contexts.

Another important limitation is the potential risk that open-sourcing MAD-Sherlock might allow adversaries to train models specifically designed to counter or evade detection by our system. As adversarial actors gain access to the source code, they could exploit its known strengths and weaknesses to develop countermeasures that diminish its effectiveness. However, despite these risks, we believe that open-sourcing remains the right path forward. Open-sourcing encourages transparency, collaboration, and rapid innovation, enabling the broader community to contribute improvements, detect vulnerabilities, and build on the system.

Moreover, by engaging the community, we can foster the development of more resilient and adaptive models that evolve in response to emerging adversarial techniques, thus maintaining MAD-Sherlock's effectiveness in the long term. The collective strength of a diverse, open-source community can outweigh the potential threats posed by adversarial exploitation.

Future work will need to address these limitations to enhance the practical utility, robustness, and long-term resilience of MAD-Sherlock.

## A.2 Sample Image-Caption Pair in the News Domain

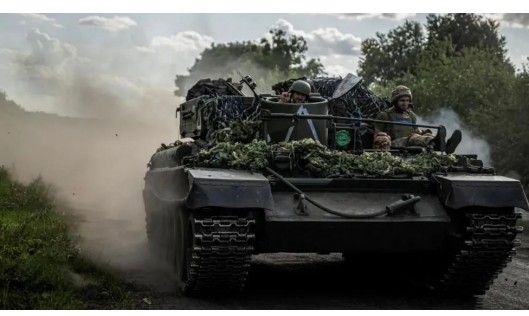

Figure 4: Russian President Vladimir Putin has called Ukraine's move into Kursk a "major provocation". Image and caption taken from the BBC article here (Accessed at 17:43 on Aug 11, 2024): `https://www.bbc.co.uk/news/articles/cze5pkg5jwlo`

## A.3 Prompts for MAD-Sherlock

```
This is a summary of news articles related to the image: {}
Based on this, you need to decide if the caption given below
belongs to the image or if it is being used to spread false
information to mislead people.
CAPTION: {}
Note that the image is real.  It has not been digitally altered.
Carefully examine the image for any known entities, people,
watermarks, dates, landmarks, flags, text, logos and other
details which could give you important information to better
explain your answer.
The goal is to correctly identify if this image caption pair is
misinformation or not and to explain your answer in detail.
At the end give a definite YES or NO answer to this question:
IS THIS MISINFORMATION?
```

Figure 5: Initial prompt for independent opinion formation and response generation

```
This is what I think:  {}.
Do you agree with me?
If you think I am wrong then convince me why you are correct.
Clearly state your reasoning and tell me if I am missing out on
some important information or am making some logical error.
Do not describe the image.
At the end give a definite YES or NO answer to this question:
IS THIS MISINFORMATION?
```

Figure 6: Prompt for Debate Round 1

```
I see what you mean and this is what I think:  {}.
Do you agree with me?
If not then point out the inconsistencies in my argument (e.g.
location, time or person related logical confusion) and explain
why you are correct.
If you disagree with me then clearly state why and what
information I am overlooking.
Find disambiguation in my answer if any and ask questions to
resolve them.
I want you to help me improve my argument and explanation.
Don't give up your original opinion without clear reasons, DO NOT
simply agree with me without proper reasoning.
At the end give a definite YES or NO answer to this question:
IS THIS MISINFORMATION?
```

Figure 7: Prompt for Debate after Round 1

### A.4 USER STUDY

We conduct a user study to assess the effectiveness of our model in detecting and explaining misinformation. Through this study, we aim to assess the persuasiveness of our system.

#### A.4.1 SETUP

The user study was designed to evaluate the effectiveness of our system in detecting and explaining misinformation. While it is easy to quantify model performance in terms of misinformation detection, there are no effective metrics to assess the quality of the explanations generated by the model. Therefore, in order to perform a thorough analysis of the system performance, a user study is essential.

A total of 30 participants volunteered to participate in this study. The group of individuals included journalists from BBC as well as students and professors from the University of Oxford. Participation was completely voluntary and no personal information was used for the purpose of analysis in this study. For a deeper analysis we further grouped the participants based on their profession into three groups, namely: Journalists, AI Academics and Others. The 'others' category included anyone who did not belong to the first two groups. The study was conducted through a Microsoft Form. Participants were shown 10 image-text pairs and were asked to decide if the image and caption when considered together was misinformation or not. They were also asked to provide a confidence rating for their answer on a scale of 0-10, with 10 being the highest confidence level. For each image-text pair, after the participants provided their initial answers, they were shown AI insights about the same image-text pair. These AI insights were the final outputs from MAD-Sherlock. Participants were then asked to reconsider their answer and again decide if the image-text pair was misinformation or not, in light of the new information from the AI agent. Participants were also required to re-evaluate their confidence score in this new answer. While it is not entirely avoidable, we did ask participants to keep aside their personal opinions of AI and consider all AI insights objectively. Participants were not allowed to access the Internet. This was done to ensure an unbiased estimate of average human performance.

The image-text pairs to include in the study were taken from the NewsCLIPpings (Luo et al. (2021)) dataset. AI insights were taken from our best-performing setup involving the GPT-4o model. Of the 10 image-text pairs presented to the participants in the study, there were 5 instances of misinformation and 5 instances of true information. Further, all model insights were true except two of them. Therefore the model accuracy for the task was 80% and we use this as the baseline accuracy to compare human performance against.

We analyse two special cases, where MAD-Sherlock argues for the wrong answer. We include these results in order to observe how persuasive our system can be even when it is wrong. We note in the instance where the image-text pair was actually misinformation and the model argued that it was not, 6 participants changed their correct responses to those suggested by MAD-Sherlock. Although this is only 5% of the participants, it still gives a significant insight into how persuasive the model

can appear even when it is wrong. While the case of false negatives is important, false positives are an even more concerning matter for our problem statement. In the case where MAD-Sherlock declared the given image-text pair to be misinformation when it was not, is important to analyse. In this setting 50% of the total participants changed their answer to the wrong one, therefore believing a piece of true information to be false. In some cases where participants chose the wrong response to begin with, their confidence in the response further increased after considering insights from the system. Finally, 4 participants did not change their answer to the wrong one after considering AI insights but their confidence in their response decreased.

The average time taken to complete the study was 12 minutes and 57 seconds. The average participant was therefore able to go through 10 image-text pairs and decide if they were misinformation or not in under 13 minutes. The same task without AI insights would require extensive analysis and we project it would take between 30-45 minutes to decide if 10 image-text pairs were misinformation.

### A.5  MULTI-MODAL DEBATES FOR HAMFUL MEME DETECTION

While this work relates to a different problem than OOC misinformation detection in the news domain, we still find the approach taken by the authors a relevant related work and therefore include it here. Lin et al. (2024) use LMMs debating against each other to generate explanations for contradictory arguments regarding whether a given meme is harmful. These explanations are then used to train a small language model as a judge to determine whether the image and text that make up the meme are actually harmful. This work does not allow agents to have flexibility of opinion. There are always two agents, and each one is provided a stance to defend. Moreover, a judge decides the final outcome of the debate and needs to be trained on data from the debate. This method also does not benefit from external retrieval, and therefore, the debating agents are not aware of the crucial external context related to the input. Finally, this work is related to harmful *meme* detection and does not concern the problem of misinformation detection in the news domain, which likely requires more intricate contextual analysis, including of external context.

