# OpenReview forum: "MAD-Sherlock: Multi-Agent Debates for Out-of-Context Misinformation Detection"
_ICLR.cc/2025/Conference — Submitted to ICLR 2025_

### Official Review · Reviewer_AktV · 2024-10-27

**Soundness:** 3
**Presentation:** 3
**Contribution:** 3
**Rating:** 6
**Confidence:** 4

**Summary:**

This paper presents MAD-Sherlock, a multi-agent debate system designed for out-of-context misinformation detection. MAD-Sherlock leverages a multi-agent framework where each agent independently analyzes image-text pairs and engages in multiple rounds of debate to assess contextual consistency. The system incorporates external information retrieval to enhance the agents' reasoning capabilities, achieving high detection accuracy and explainability without the need for task-specific fine-tuning. A unique aspect of MAD-Sherlock is its systematic construction and comparison of various multi-agent debate strategies, offering a comprehensive exploration of debate structures within a multi-agent framework for OOC detection.

**Strengths:**

1. This paper applies external information retrieval and multi-agent collaboration to the task of out-of-context misinformation detection, which is a relatively novel approach. The authors effectively combine these elements in this context, and the experimental results demonstrate the effectiveness of this method.

2. Unlike previous work, MAD-Sherlock specifically constructs and compares different multi-agent debate strategies, providing a systematic analysis of various debate methods in out-of-context detection. This exploration is quite interesting, and it also offers valuable insights for applying multi-agent frameworks in similar tasks.

3. The authors built a complete pipeline that combines image and text processing, including external data collection and cleaning, which is a substantial effort. Handling multimodal data and integrating external information adds complexity to the system.

**Weaknesses:**

1. Multi-agent collaboration and debate strategies are popular methods for improving results, and incorporating external information is also common in multi-agent setups like AutoGPT. The experiments here don’t include comparisons with these popular frameworks, especially those that also use external data and agent collaboration. Adding such comparisons would highlight where MAD-Sherlock stands out.

2. While external retrieval improves the system’s accuracy, it could backfire if relevant information isn’t available or if the search results are inconsistent or irrelevant. The paper doesn’t delve much into this issue; it would be helpful to discuss how the system performs in cases where external information is incomplete or unavailable to gauge robustness.

3. The multi-agent debate structure adds a lot of computational cost, which isn’t fully detailed in the paper. It would be useful to see ablation studies comparing debate length and performance to understand the trade-offs in runtime and accuracy. This would make it easier to evaluate MAD-Sherlock’s feasibility for practical applications.

**Questions:**

Why did you only compare with pretrained multimodal baseline methods? Is it because this approach is more commonly used for this task? Additionally, why didn’t you include comparisons with systems that use multi-agent collaboration with external information retrieval?

---

> ### Author Response · Authors · 2024-11-28
> **Rebuttals to Weaknesses**
>
> We thank the reviewer for taking the time to thoroughly understand and review our work. We are happy to address their concerns below.
>
> ## Weaknesses
>
> **Weakness-1**: _"Multi-agent collaboration and debate strategies are popular methods for improving results, and incorporating external information is also common in multi-agent setups like AutoGPT. The experiments here don’t include comparisons with these popular frameworks, especially those that also use external data and agent collaboration. Adding such comparisons would highlight where MAD-Sherlock stands out."_
>
> We appreciate the reviewer highlighting this point. While AutoGPT and similar frameworks are powerful tools that incorporate external data and agent collaboration, they typically rely on a significantly higher number of external API calls and focus more on retrieval tasks. In contrast, MAD-Sherlock's focus is primarily on reasoning and misinformation detection, which necessitates a different approach.
>
> That said, we recognize the value of benchmarking MAD-Sherlock against these popular frameworks to understand its relative strengths and limitations better. We will explore incorporating such comparisons in the camera-ready version of the paper to further contextualize MAD-Sherlock's performance and highlight its distinct advantages.
>
> **Weakness-2**: _"While external retrieval improves the system’s accuracy, it could backfire if relevant information isn’t available or if the search results are inconsistent or irrelevant. The paper doesn’t delve much into this issue; it would be helpful to discuss how the system performs in cases where external information is incomplete or unavailable to gauge robustness."_
>
> The reviewer raises an important point regarding the reliance on external retrieval. In cases where relevant external information is incomplete or unavailable, our system is designed to proceed using the available inputs without external augmentation. During the initial phases of our work, we conducted preliminary experiments to establish that the inclusion of external information significantly enhances model performance. However, these experiments were not included in the current submission.
>
> We agree that a more thorough investigation into how the system performs under scenarios of incomplete or unavailable external information would provide valuable insights into its robustness. We plan to include these experiments in the camera-ready version, along with a detailed discussion of their implications. We thank the reviewer for bringing this to our attention.
>
> **Weakness-3**: _"The multi-agent debate structure adds a lot of computational cost, which isn’t fully detailed in the paper. It would be useful to see ablation studies comparing debate length and performance to understand the trade-offs in runtime and accuracy. This would make it easier to evaluate MAD-Sherlock’s feasibility for practical applications."_
>
> We acknowledge that the multi-agent debate structure introduces additional computational cost. To address this, we will include detailed cost and latency analyses in our revised submission. These will quantify the trade-offs between debate length, runtime, and performance, helping to evaluate the feasibility of MAD-Sherlock for various practical applications.
>
> It is also worth noting that the framework itself is modular, allowing the underlying models to be replaced with smaller, more efficient ones for scenarios where computational resources are constrained. While there is a trade-off between performance and computational cost, we believe the significant performance improvements observed in our experiments justify the additional cost. However, this trade-off may vary depending on the use case. We have included information related to the time and cost efficiency of our method in our updated submission.

---

> > ### Comment · Reviewer_AktV · 2024-12-02
> >
> > Thank you for your response. I am happy to increase the score if you include those plans in the updated submission. However, as they remain just a plan, I will maintain my current score. Additionally, for the cost, I believe a comparison is needed rather than focusing solely on your method.

---

> ### Author Response · Authors · 2024-11-28
> **Rebuttals to Questions**
>
> ## Questions
>
> **Q1**: _Why did you only compare with pretrained multimodal baseline methods? Is it because this approach is more commonly used for this task? Additionally, why didn’t you include comparisons with systems that use multi-agent collaboration with external information retrieval?_
>
> We appreciate the reviewer’s question. Our selection of baseline methods was guided by the specific requirements of our problem statement, which focuses on determining whether an image-text pair constitutes misinformation. For this task, multimodal reasoning is essential, as it directly incorporates both image and text inputs rather than relying solely on text-based descriptions of images.
>
> We initially experimented with text only models where input was a textual description of the image and the corresponding text but found the results to be suboptimal. These experiments established that a multimodal approach is critical for effectively addressing this task.
>
> We recognize the value of including multi-agent systems that incorporate external information retrieval, as suggested. This will be explored in future work and has been mentioned as a potential direction for enhancing our comparative framework in our updated submission.

---

### Official Review · Reviewer_UiNZ · 2024-10-29

**Soundness:** 2
**Presentation:** 3
**Contribution:** 2
**Rating:** 5
**Confidence:** 4

**Summary:**

MAD-Sherlock is a multi-agent debate system designed to detect out-of-context misinformation by analyzing inconsistencies between images and accompanying text. Unlike traditional AI models, it enables multiple multimodal agents to independently assess and debate the context of information, using external retrieval to enhance accuracy and provide clear, explainable insights.

**Strengths:**

1. The MAD-Sherlock framework introduces a multi-agent debate approach to detect out-of-context misinformation, combining asynchronous debates with external information retrieval to enhance the model's contextual understanding and interpretability. This approach shows significant innovation compared to single-agent methods.
2. The model provides human-readable explanations during the decision-making process, which is a major improvement over current AI-driven misinformation detection systems.
3. This work attempts to leverage internet searches to extract external information and use it to enhance the performance of misinformation detection and proves its effectiveness.
5. Interesting findings:
    1. The comparison of various debate methods reveals that asynchronous debate is the most effective, providing valuable insights for designing multi-agent debate frameworks.
    2. There is a significant performance improvement when models believe they are debating against a human rather than another AI agent.
    3. The method also allows agents the freedom to change their opinions mid-debate. In such settings, agents demonstrate an enhanced ability to critically evaluate arguments and identify subtle inconsistencies.

**Weaknesses:**

1: Lack of data cleaning for external information

Apart from what was mentioned in Appendix A.1:

> First, while our model excels at detecting out-of-context image-text pairs, its reliance on external retrieval can lead to reduced accuracy when relevant context is unavailable or difficult to retrieve.

(1) The paper seems to lack verification of the authenticity and quality of the extracted external information. Without thorough data cleaning, if low-quality data or even fake news is retrieved, it could negatively impact the judgment results.

(2) During the pre-training process, commercial LLMs use carefully cleaned data. If a conflict arises between the parameter knowledge of the LLM itself and the external knowledge, how should it be resolved? This is not uncommon; when an event occurs, it is often accompanied by numerous rumors, even conflicting ones. Blindly trusting the external knowledge retrieved online could lead to undesirable outcomes.

(3) Only the Bing Visual Search API was used for information retrieval. Is it proven to be reliable and effective enough?

2: Reliability and effectiveness of LLM summarization, and potential side effects

In Section 3.3.2, you mentioned using LLMs, such as Llama-13B, to summarize information, focusing only on the most important parts of the text. I am curious about how it determines the most important parts and whether it might miss important details. Could the performance of Llama-13B itself become a bottleneck in the workflow?

Information summarized and rewritten by the LLM inevitably alters the original language pattern, and when such processed information is provided to the agents, could it make the already potentially unreliable external information even harder to detect?

3: Limited dataset

Only the NewsCLIPpings dataset was used, which may lack representativeness. This dataset is from 2021, a time when LLMs were not as prevalent as they are now, and AIGC content was limited. I question whether it is representative of the current and future online news landscape and the ability of this work to detect LLM-generated misinformation.

4: Questionable fairness of the user study

As mentioned in Appendix A.4.1, in the user study, participants were not allowed to access the internet and could not retrieve external information (e.g., Bing Visual Search API) like MAD-Sherlock. They could only rely on their own experience and common ensense, which is unfair. At the very least, a control group should be added, allowing participants to access the same external information as MAD-Sherlock.

**Questions:**

see above.

---

> ### Author Response · Authors · 2024-11-28
> **Rebuttals to Weaknesses**
>
> We thank the reviewer for their time and efforts and address their concerns below.
>
> ## Weaknesses
>
> **Weakness-1**: _"Lack of data cleaning for external information"_
>
> **Weakness-1.1**:  _"The paper seems to lack verification of the authenticity and quality of the extracted external information. Without thorough data cleaning, if low-quality data or even fake news is retrieved, it could negatively impact the judgment results."_
>
> We thank the reviewer for bringing up a crucial point related to the quality of the retrieved external information. While we currently only opt for a qualitative analysis of the retrieved information due to the large scale of the NewsCLIPpings dataset, we would like to include some form of quantitative analysis of the retrieved information as well.
>
> We mitigate the risk of retrieving and relying solely on fake news by aggregating information from multiple independent webpages to construct the external context. This approach ensures a more diverse and balanced set of sources, significantly reducing the likelihood that the retrieved context is dominated by misinformation and thereby minimizing its potential impact on the results.
>
> **Weakness-1.2**: _"During the pre-training process, commercial LLMs use carefully cleaned data. If a conflict arises between the parameter knowledge of the LLM itself and the external knowledge, how should it be resolved? This is not uncommon; when an event occurs, it is often accompanied by numerous rumors, even conflicting ones. Blindly trusting the external knowledge retrieved online could lead to undesirable outcomes."_
>
> This is also addressed with the previous point.
>
> **Weakness-1.3**: _"Only the Bing Visual Search API was used for information retrieval. Is it proven to be reliable and effective enough?"_
>
> We explored using the Google Visual Search API but found that it did not support reverse image-based search, which was a critical requirement for our use case. Given the limited availability of visual search APIs, we turned to the Bing API, which offers robust access to a substantial pool of internet resources which is crucial to making informed decisions about the authenticity of the image and text input. We are open to suggestions by the Reviewer as to what additional data sources could be integrated into our approach.
>
> **Weakness-2**: _"Reliability and effectiveness of LLM summarization, and potential side effects
> In Section 3.3.2, you mentioned using LLMs, such as Llama-13B, to summarize information, focusing only on the most important parts of the text. I am curious about how it determines the most important parts and whether it might miss important details. Could the performance of Llama-13B itself become a bottleneck in the workflow?
> Information summarized and rewritten by the LLM inevitably alters the original language pattern, and when such processed information is provided to the agents, could it make the already potentially unreliable external information even harder to detect?"_
>
> The model determines the most important parts of the text only based on the prompt that is provided to it. The model is specifically prompted to summarize the given text based on the most important parts of the input. We also prompt the model to only base its output on the input text and not introduce any new information into the generated summary. We acknowledge the reviewer’s concern that the performance of the Llama-13B model could possibly become a bottleneck and we are happy to include ablation studies with a better summarization model with the CRC. However, random manual qualitative checks of the generated summaries do not indicate that summarization lead to a general loss of information. Leveraging more advanced and refined models like Llama3, we believe, would only further improve system performance. We also include this as a part of our future work section.That being said, we agree with the reviewer and acknowledge that there is a definite tradeoff between the computational efficiency of the summarization model and the quality of the summaries and in turn our system performance which should be considered based on the criticality of the use-case where MAD-Sherlock is being used. However with the current setup it should be straightforward to replace the llama-13B model with a larger/more powerful or smaller/less powerful model.
>
> We acknowledge that leveraging an LLM for generating summaries introduces a potential risk of incorporating unreliable or false elements, which could compromise the reliability of the external information. However, this risk is a general limitation of LLMs rather than a specific issue with our approach. Furthermore, we have not found any evidence to suggest that this has been an issue in practice. We believe that using more aligned and safer models for this task in the future could further mitigate this risk.

---

> ### Author Response · Authors · 2024-11-28
> **Rebuttals to Weaknesses (continued) and Final Words**
>
> **Weakness-3**: _"Limited dataset: Only the NewsCLIPpings dataset was used, which may lack representativeness. This dataset is from 2021, a time when LLMs were not as prevalent as they are now, and AIGC content was limited. I question whether it is representative of the current and future online news landscape and the ability of this work to detect LLM-generated misinformation."_
>
> We acknowledge this limitation and appreciate the reviewer’s observation. However, the NewsCLIPpings dataset remains one of the largest and most widely used datasets for fake news detection, serving as a community-accepted benchmark. For this reason, we believe it is essential to demonstrate our results on this dataset to ensure a fair and meaningful comparison with existing methods.
>
> That said, we fully agree with the reviewer that the dataset is outdated, particularly given the evolving prevalence of LLM-generated content and already include the need for a continual more up-to-date dataset as a part of future work. Additionally, in the external retrieval component of our approach, we observed that some of the webpages linked to the dataset samples are no longer accessible due to the dataset's age. While the number of unavailable webpages is currently not significant enough to impact our results, it highlights the importance of transitioning to more recent and relevant datasets.
>
> **Weakness-4**: _"Questionable fairness of the user study: As mentioned in Appendix A.4.1, in the user study, participants were not allowed to access the internet and could not retrieve external information (e.g., Bing Visual Search API) like MAD-Sherlock. They could only rely on their own experience and common ensense, which is unfair. At the very least, a control group should be added, allowing participants to access the same external information as MAD-Sherlock."_
>
> We agree that our current findings can definitely be refined by using a more refined user study. We are in the process of conducting the study and would like to include the results in the camera ready version of the paper. However, on the fairness of our study we would like to clarify that our study is motivated by our concern about how informative and trustworthy MAD-Sherlock’s insights are for humans. We wanted to understand if the insights confuse human participants or further enhance their line of reasoning. In this set-up, we believe internet access could be considered a confounding factor in the study as it introduces variability in participants' ability to search, interpret, and evaluate online content. By restricting internet access, we aimed to isolate the effectiveness of MAD-Sherlock's insights without external influences, ensuring that the results reflect the system's intrinsic utility rather than participants' independent research skills. In summary, we would like to clarify that our empirical results do not indicate that MAD-Sherlock alone can outperform human experts with access to the internet and without time constraints. Rather, our results indicate that MAD-Sherlock can help journalists make better decisions under time constraints, and that it can significantly uplift unskilled humans, e.g. in a citizen intelligence context.
>
> ## Final Words
> We would like to thank the reviewer for their time once again. We hope that our answers help clarify their concerns and the reviewer might consider increasing their score.

---

### Official Review · Reviewer_peoS · 2024-10-31

**Soundness:** 2
**Presentation:** 2
**Contribution:** 2
**Rating:** 5
**Confidence:** 5

**Summary:**

MAD-Sherlock, a Multi-Agent Debate system for detecting out-of-context (OOC) misinformation, addresses issues of existing AI detection systems. It introduces a novel framework where multimodal agents collaborate to assess context consistency and request external info. Enables explainable detection with high accuracy without domain-specific fine-tuning. The experimental results confirm external retrieval improves accuracy, and user studies show it boosts performance for both experts and non-experts, making it a powerful tool for intelligence applications.

**Strengths:**

* The paper presents a meaningful problem in multimodal fake content detection: OOC (Out-of-Context).
* The method proposed in the paper is very interesting and achieved state-of-the-art (SOTA) results, validating its effectiveness.
* The paper is written in great detail.

**Weaknesses:**

* The paper primarily addresses the Out-of-Context (OOC) issue of fake online content. However, it does not provide a detailed explanation or analysis of why the debate approach was introduced and how it effectively addresses the OOC problem.

* The introduction of external information retrieval can lead to label leakage issues.

* The selection of baselines in the paper is not enough; it should include some multimodal fake online content detection methods[1,2] as baselines for comparison. For example, models like GPT-4o were not designed specifically for fake online content detection, so the comparison methods in the paper lack convincing power.

* This paper lacks sufficient ablation experiments to demonstrate the effectiveness of each component of MAD-Sherlock and its contribution to the overall performance.

* In the section 4.3.1,

>We also observe a significant performance increase when the agent believes it is conversing with a human instead of another AI agent

the paper lacks an explanation and analysis to clarify why this phenomenon occurs.


So, I think this paper needs further work.

References:

[1].Chen Y, Li D, Zhang P, et al. Cross-modal ambiguity learning for multimodal fake news detection[C]//Proceedings of the ACM web conference 2022. 2022: 2897-2905.

[2]. Qian S, Wang J, Hu J, et al. Hierarchical multi-modal contextual attention network for fake news detection[C]//Proceedings of the 44th international ACM SIGIR conference on research and development in information retrieval. 2021: 153-162.

**Questions:**

1. Why introduce the debate framework to address the OOC problem? It would be helpful if the authors could clarify the insight behind this choice.

2. For the different debate strategies, how do they vary in addressing the OOC problem? Apart from the results shown in Table 1, is there additional analysis or explanation provided here?

3. How can it be ensured that the introduction of external information retrieval will not lead to label leakage issues?

Additionally, it would be helpful if the authors could address the issues mentioned in the "weaknesses" section.

---

> ### Author Response · Authors · 2024-11-28
> **Rebuttals to Weaknesses**
>
> We thank the reviewer for taking the time to carefully consider our paper. We are happy to address their concerns and add corresponding improvements to our paper.
>
> ## Weaknesses
> **Weakness-1**: _"The paper primarily addresses the Out-of-Context (OOC) issue of fake online content. However, it does not provide a detailed explanation or analysis of why the debate approach was introduced and how it effectively addresses the OOC problem."_
>
> Firstly, we opt for a multiagent setup to allow for clean separation of agent contexts and decentralisation of action spaces. In addition to this, the problem of OOC misinformation detection requires looking at the input from multiple perspectives which is something a single model is less equipped to do. We therefore opt for a multi-agent setup. With regard to the debate setup in particular, in our work we extend the insights from  [1], which shows that robustness and interpretability of Large Language Models (LLMs) is improved when multiple LLMs are placed in a debate environment. Our motivation to apply multi-agent debate to OOC misinformation detection is also partially motivated by the successful application of such settings to reasoning tasks, which we believe are at least somewhat related to OOC detection tasks [2]. We structure the conversation between different agents as a debate to allow for difference of opinion and cleanly separates role contexts, which we find enables models to uncover different elements of misinformation.
>
> During preliminary experiments we also explored non-debate configurations and observed agents often converged rapidly to the same answers regardless of correctness. This limits the agents’ ability to look at the input from different aspects. In contrast to this, we observe that putting models in a debating setup with the ability to change line-of-reasoning and debate stance, enabled more informed and substantive discussions around the potential elements of misinformation in the input, which in turn helped improve performance and explainability. We can include experimental evidence on a smaller subset of the data to further support these claims.
>
> It is also important to address how “debates” are defined in our work, which we should include in the paper as well. Agents are not provided predefined stances and are allowed to choose independent positions based on the input. Agents are further allowed to change their position mid-”debate” therefore engaging in a debate or reaching consensus. While we refer to this form of interaction between the agents as a “debate”, it diverges from the conventional setup of one.
>
> **Weakness-2**: _"The introduction of external information retrieval can lead to label leakage issues."_
>
> We appreciate the concern around label leakage issues that can arise from the external information retrieval. However, we would like to clarify that our methodology does not involve finetuning or training the models on any sort of retrieved information. The external retrieval module operates independently of model parameters which remain unchanged during our experiments. The primary focus of our work is to propose a novel approach towards explainable misinformation detection without any domain-specific finetuning. Therefore we believe that the concern around label leakage is not applicable to our setup. Our external information retrieval module is designed to provide agents with additional context related to the input and is independent of the ground truth labels.
>
> If the concern is around the test data itself, we can confirm that a thorough qualitative analysis was performed to ensure that the retrieved information does not contain the label itself and is only limited to news articles related to the image.
>
> Should there remain concerns related to specific scenarios, we are open to conducting ablation studies to explicitly establish and demonstrate the absence of label leakage issues. Could the Reviewer kindly clarify if our understanding of their concerns is correct?

---

> ### Author Response · Authors · 2024-11-28
> **Rebuttals to Weaknesses (continued)**
>
> **Weakness-3**: _"The selection of baselines in the paper is not enough; it should include some multimodal fake online content detection methods[1,2] as baselines for comparison. For example, models like GPT-4o were not designed specifically for fake online content detection, so the comparison methods in the paper lack convincing power."_
>
> The baselines selected for our work represent a comprehensive range of state-of-the-art methods in the field, encompassing diverse approaches from simple MLP fine-tuning models to advanced multimodal reasoning frameworks. These baselines were carefully chosen to provide a balanced and rigorous evaluation of our method’s performance against both foundational and cutting-edge techniques.
> While the works suggested by the reviewer are interesting, we do not find them directly comparable to our method. Specifically, the suggested methods target broader multimodal tasks or are optimized for use cases distinct from the detection of fake online content. Including such baselines, while informative, would not provide an equitable comparison given the methodological and task-specific differences.
> While both MAD-Sherlock and COOLANT [3] address multimodal fake news detection, their objectives, methodologies, and scopes differ fundamentally, making direct comparisons unsuitable. MAD-Sherlock focuses on misinformation detection through multi-agent debate strategies that simulate human reasoning, emphasizing explainability and the integration of external contextual information. In contrast, COOLANT optimizes feature alignment and aggregation using cross-modal contrastive learning within a dual-encoder framework, prioritizing classification accuracy rather than reasoning or contextual adaptability. Furthermore, MAD-Sherlock evaluates on the NewsCLIPpings dataset with an emphasis on reasoning under complex misinformation scenarios, while COOLANT is tailored to social media datasets like Twitter and Weibo, focusing on alignment-based classification tasks. These distinctions underline that the two approaches address different aspects of multimodal fake news detection, making direct comparisons impractical. In addition to this, COOLANT also suffers from the same short-comings as the other, possibly more relevant, methods we compare against: 1. It lacks the essential component of reproducibility, 2. It requires extensive finetuning, while MAD-Sherlock does not require any fine-tuning.
>
> The second suggested work (HMCAN) [4] employs a methodology from 2021, using ResNet for image feature extraction and BERT for textual feature extraction, which are then fused through a multi-modal contextual attention network. While this approach was notable at the time, our paper already includes comparisons with more advanced and contemporary methods that better align with the current state of the field. Additionally, HMCAN is limited to providing binary classification scores and does not address explainability, which is a core focus of our proposed system.
>
> Also both of the above works, might be more inclined towards Chinese content and fake news detection, which is not supported by our system. We currently are unable to use datasets that have been created using content from websites in languages other than English since our external retrieval module only supports English, as well as the lack of multilingual datasets. As an extension of the project we would like to add multilingual capabilities to the system and have included it as one of the future works. We believe that preliminary multilingual capabilities can be added through in-context instructions, although support for minority languages may be harder to attain - which is, of course, a reflection of systemic issues, and not our specific research project.
>
> If the reviewer is still concerned about our selection of baseline methods, we would be open to adding more related methods for comparison in our camera ready submission as long as:
> those methods have not been shown to be outperformed by our baselines, and,
> the methods are amenable to explainability.
>
> **Weakness-4**: _"This paper lacks sufficient ablation experiments to demonstrate the effectiveness of each component of MAD-Sherlock and its contribution to the overall performance."_
>
> We agree that this would be a valuable addition to the paper and would be including this in our camera ready submission. We thank the reviewer for bringing this to our attention.

---

> ### Author Response · Authors · 2024-11-28
> **Rebuttals to Weaknesses (continued) and References**
>
> **Weakness-5**: _the paper lacks an explanation and analysis to clarify why performance increases when the agent believes it is conversing with a human instead of another AI agent_
>
> We acknowledge the reviewer’s observation and agree that this phenomenon requires further investigation. While we do not have a definitive explanation for why this occurs, we propose a few potential hypotheses which could potentially inform future work in this direction:
>
> 1. Training data: a substantial portion of the data that large models are trained on is human-generated content, which may implicitly condition the model to respond differently or more robustly to a potential human compared to another agent.
> 2. Agents reward heuristics during inference: the agent could internally optimize for human-centric conversations/interactions which could explain better performance when it believes a human is part of the exchange.
> 3. Training process: the training process for many large language models also involves reward formulation based on human preferences and feedback which could also lead to the development of an implicit bias in the model towards interactions involving humans.
> 4. Commercial LLMs are additionally safety-finetuned. This safety-finetuning may prompt the model to behave differently in the context of human users or contexts.
>
> While these are speculative experiments, the observation is consistent across all our experiments. We believe this can provide a valuable insight for future work related to designing interaction configurations for models to improve performance of multi-agent systems.
>
> ## References
> [1] Akbir Khan, John Hughes, Dan Valentine, Laura Ruis, Kshitij Sachan, Ansh
> Radhakrishnan, Edward Grefenstette, Samuel R. Bowman, Tim Rocktäschel,
> and Ethan Perez. Debating with more persuasive llms leads to more truthful
> answers, 2024.
>
> [2] Haotian Wang, Xiyuan Du, Weijiang Yu, Qianglong Chen, Kun Zhu, Zheng Chu, Lian Yan and  Yi Guan. Learning to break: knowledge-enhanced reasoning in multi-agent debate system, 2023.
>
> [3].Chen Y, Li D, Zhang P, et al. Cross-modal ambiguity learning for multimodal fake news detection[C]//Proceedings of the ACM web conference 2022. 2022: 2897-2905.
>
> [4]. Qian S, Wang J, Hu J, et al. Hierarchical multi-modal contextual attention network for fake news detection[C]//Proceedings of the 44th international ACM SIGIR conference on research and development in information retrieval. 2021: 153-162.

---

> ### Author Response · Authors · 2024-11-28
> **Rebuttals to Questions and Final Words**
>
> ## Questions
>
> **Q1**: _Why introduce the debate framework to address the OOC problem? It would be helpful if the authors could clarify the insight behind this choice._
>
> Addressed under rebuttal to weaknesses.
>
> **Q2**: _For the different debate strategies, how do they vary in addressing the OOC problem? Apart from the results shown in Table 1, is there additional analysis or explanation provided here?_
>
> The debating strategies were part of a preliminary set of experiments to determine which one would be best suited for extensive experimentation going forward. The main objective behind trying different debating strategies was not to directly detect OOC but to see which interaction configuration enabled the most substantial discussions and allowed for better explainability. We agree that this should be further clarified in section 3.1 to avoid possible confusion.
>
> **Q3**: _How can it be ensured that the introduction of external information retrieval will not lead to label leakage issues?_
>
> Addressed under rebuttal to weaknesses.
>
> ## Final Words
> Once again, we thank the reviewer for their valuable insights and feedback. We hope we have sufficiently addressed their concerns and the reviewer would consider increasing their score.

---

> > ### Author Response · Authors · 2024-12-02
> > **Did we address your concerns?**
> >
> > Dear Reviewer peoS,
> >
> > As the rebuttal period is coming to a close, we would like to ask whether we have successfully addressed your concerns, or whether there is anything else that you would like to see addressed.
> >
> > In particular, please note the additional empirical baseline clarifying the utility of the debate approach, and our responses to your other concerns about relevant baselines.
> >
> > Many thanks
> >
> > The Authors

---

> > ### Comment · Reviewer_peoS · 2024-12-03
> >
> > Thank you very much for your response! Your reply addressed most of my concerns, and I will increase my score. However, the response to "Weakness-1" did not convince me. I believe your explanation is more about why the debate approach is used to address misinformation rather than solving OOC, so the motivation here is not very clear. Also, regarding "Weakness-4," which mentions ablation experiments, I did not see sufficient results from the ablation experiments. Therefore, I will increase my score by 2 points.
> >
> > Lastly, I really appreciate the author's effort and time!

---

### Official Review · Reviewer_xyn6 · 2024-11-04

**Soundness:** 3
**Presentation:** 3
**Contribution:** 3
**Rating:** 6
**Confidence:** 3

**Summary:**

The paper describes a method to detect a particular kind of misinformation, where text is paired with an image misleadingly out of context, using debating LLMs. The LLMs are equipped with a reverse image web search retrieval system. The paper shows this system performs well compared to many baselines and alternatives from the literature, as well as in a user study assessing how much the explanations the LLMs generate help humans.

**Strengths:**

Important problem and the general approach of retrieval-augmented LLMs, with something to improve their reasoning (here, debate), makes a lot of sense.

Comparison with many alternative methods from literature.

Good experiments on different debate setups, both as an ablation here and potentially informative for methods in other domains too.

**Weaknesses:**

A baseline with the LLM and retrieval system used but without debate - similar to actor-skeptic but without the skeptic - seems missing. I feel like this is important to understand how much debate is actually helping, since it isn't guaranteed that it would perform worse than e.g. some less effective forms of debate that might confuse things more, or compared to other methods from the literature which might be using models weaker than GPT-4o.

Cost and time efficiency are not reported. This also connects to the previous point, and seems a key consideration when comparing multiple LLMs engaging in multi-turn debates, which could be significantly more costly than e.g. a single LLM setup. A high cost could be an important limitation, and regardless, important information for readers considering if they could apply the work.

Although - aside from the baseline point mentioned above - the comparisons with existing methods are extensive, they are all performed on a single dataset. The margin compared to the next-best performing approach (Sniffer with fine-tuning) is only about 1.7%, and there are no error bars reported. So, it's not very clear how definitive the performance conclusions are.

Overall, the combination of the three preceding two points forms my main concern: the framework looks promising, but some information is missing for a reader to make a full, confident assessment. Below I note two minor issues I had with the writing:

Discussion of Lin et al (line 167): it's clear the current work is quite different. It's less clear to me, though, why this work is highlighted in general, given that it is so different, including entirely different domain. Maybe this could be contextualized a bit more broadly in terms of approaches to classification by debating LLMs, or some other connecting insight or argument beyond "here's another work that used debating LLMs".

Section 3.1: I was a bit unclear when first reading this on what is background information on possible ways a debate could be structured, vs. what you actually test yourselves. Maybe the wording could be a bit more explicit that you test all of these.

**Questions:**

Why summarize using Llama 13B as opposed to a more recent Llama? It seems like Llama 3 8b is both smaller and has significantly better performance?

The user study asks participants to not search the web themselves. I can see that being applicable for laypeople, who might not want to spend time checking stuff or know what should be checked. I'm less sure for journalists, are there cases where they too wouldn't be using web search?

---

> ### Author Response · Authors · 2024-11-28
> **Rebuttals to Weaknesses**
>
> We thank the reviewer for their feedback and address their concerns below.
>
> ## Weaknesses
>
> **Weakness-1**: _"A baseline with the LLM and retrieval system used but without debate - similar to actor-skeptic but without the skeptic - seems missing. I feel like this is important to understand how much debate is actually helping, since it isn't guaranteed that it would perform worse than e.g. some less effective forms of debate that might confuse things more, or compared to other methods from the literature which might be using models weaker than GPT-4o."_
>
> We agree that including a single-agent baseline enhances comparison and performance analysis. Accordingly, we have included preliminary but statistically significant results for the requested single-agent baseline on a randomly sampled subset (10% of the full dataset) in our updated submission. Results on the entire dataset will be included in the camera-ready version.
> Our findings confirm that the debate setup significantly outperforms the single-agent setup, providing strong evidence for our hypothesis that multi-agent debate offers inherent advantages by leveraging separate context windows (as suggested in prior work, e.g. https://arxiv.org/abs/2305.14325). Furthermore, our qualitative analysis highlights that the debate setup improves explainability, as distinct context windows allow for better role-specific separation. This critical insight has been added to the updated submission.
>
> **Weakness-2**: _"Cost and time efficiency are not reported. This also connects to the previous point, and seems a key consideration when comparing multiple LLMs engaging in multi-turn debates, which could be significantly more costly than e.g. a single LLM setup. A high cost could be an important limitation, and regardless, important information for readers considering if they could apply the work."_
>
> We appreciate the reviewer’s suggestion to include details on cost and time efficiency, which we have added to the updated submission. Our approach avoids finetuning, making it significantly more cost-effective compared to prior methods reliant on extensive finetuning. While we report results with a more powerful model, our system is model-agnostic and can readily use any open-source alternative for greater cost and time efficiency. Given the notable performance gains, we believe the overall cost of our method is well-justified.
>
> **Weakness-3**: _"Although - aside from the baseline point mentioned above - the comparisons with existing methods are extensive, they are all performed on a single dataset. The margin compared to the next-best performing approach (Sniffer with fine-tuning) is only about 1.7%, and there are no error bars reported. So, it's not very clear how definitive the performance conclusions are."_
>
> With regard to concern around reporting results only on a single dataset, we report all results on the NewsCLIPpings dataset which is the community accepted benchmarking dataset for the task of out of context misinformation detection. We do this in order to compare to existing baseline methods. We would also like to emphasize that our proposed method does not require any finetuning compared to Sniffer and we not only significantly outperform the unfinetuned version of the model but also the finetuned version therefore achieving state of the art performance across all baselines and related methods.
>
> **Weakness-4**: _"Discussion of Lin et al (line 167): it's clear the current work is quite different. It's less clear to me, though, why this work is highlighted in general, given that it is so different, including entirely different domain. Maybe this could be contextualized a bit more broadly in terms of approaches to classification by debating LLMs, or some other connecting insight or argument beyond "here's another work that used debating LLMs"."_
>
> We also agree that this work does not directly relate to MAD-Sherlock. The mentioned work, approaches the problem of harmful meme detection using a multi-agent setup. We initially included it to provide a more comprehensive overview of related works and this work is one of the few to use debating multi-modal models. We understand it might not directly relate to our work here since the problem of misinformation detection in the news domain is significantly different from that of detecting harmful or offensive memes. We have moved this work to the appendix in our updated submission.
>
> **Weakness-5**: _"Section 3.1: I was a bit unclear when first reading this on what is background information on possible ways a debate could be structured, vs. what you actually test yourselves. Maybe the wording could be a bit more explicit that you test all of these."_
>
> We would like to thank the reviewer for bringing this to our notice. We agree that making the fact that we test all debate setups in order to select the best one more explicit would make the section more clear. We have now incorporated this into our updated submission.

---

> ### Author Response · Authors · 2024-11-28
> **Rebuttals to Questions and Final Words**
>
> ## Questions
> **Q1**: _Why summarize using Llama 13B as opposed to a more recent Llama? It seems like Llama 3 8b is both smaller and has significantly better performance?_
>
> We appreciate the reviewer’s suggestion to use a more recent model, such as Llama3-8B. We are happy to include those ablations using a more recent and better performing model in the CRC. That being said, we don’t have evidence to believe that summarization led to a loss of information on average. We also include this as a part of our future work section.
>
> **Q2**: _The user study asks participants to not search the web themselves. I can see that being applicable for laypeople, who might not want to spend time checking stuff or know what should be checked. I'm less sure for journalists, are there cases where they too wouldn't be using web search?_
>
> The primary objective of MAD-Sherlock is to offer a solution that minimizes the effort required from the end user, allowing them to verify image-caption pairs without needing to perform additional tasks such as web searches. This approach is particularly relevant for laypeople in a citizen intelligence setting, but we believe it is also applicable to journalists in certain scenarios.
>
> As we learned from our domain project partners, journalists often face tight deadlines and high workloads, where the ability to quickly assess the credibility of content is essential. By removing the need for manual web searches, MAD-Sherlock significantly reduces the time and cognitive effort required for verification. For example, in our user study, participants took less than 13 minutes on average to complete the evaluation of 10 image-caption pairs using AI-generated insights. In contrast, performing this task manually, including web searches, would have taken over 30 minutes on average. This demonstrates the potential time-saving benefits of our system, even for professionals who might have the skills and resources to perform manual verification. However, we would like to be clear that our results do not indicate that MAD-Sherlock can outperform trained human experts with full access to the internet and without time constraints.
>
> While we recognize that journalists may still choose to perform independent searches in some cases, MAD-Sherlock is designed to complement their workflows by providing immediate, actionable insights, enabling them to focus their efforts on more nuanced investigative tasks.
>
> ## Final Words
> We once again thank the reviewer for their valuable insights and feedback that helped us improve the quality of our work. We hope that our answers help further clarify the reviewer’s concerns and the reviewer would consider increasing their score.

---

> > ### Comment · Reviewer_xyn6 · 2024-12-01
> >
> > Regarding time and cost efficiency, thank you for adding the information. I would suggest, however, providing the same information for some next-best methods, such as Sniffer and GPT-4o#. The information is helpful on its own, and does not seem too exorbitant, but would be even more helpful if one could easily compare with alternatives.
> > I don't think I agree with the argument that avoiding fine-tuning reduces cost a lot, unconditionally. It reduces a potentially significant one-time cost, but if running the system on millions of examples, the inference cost may be much more relevant than the one-time fine-tuning cost. As far as I know this doesn't affect anything you've been reported in the paper, but would be careful about the argument in general.
> >
> > Table 5 of https://arxiv.org/pdf/2409.00009 suggests more powerful summarizers can have a small but possibly non-trivial effect. The setting is different, but related.
> >
> > Overall, aside from some tiny points mentioned above, the new / in progress results address most of my concerns. The main remaining one is single dataset. While this one may be the standard one, that doesn't really solve the potential issues, such as some bias or spurious correlation in the dataset that aligns better with the proposed approach than other ones. Of course, equally plausible that it could go the other way and this method would be even better on a different dataset. But still a significant element of uncertainty that affects the strength of the results.
> > Are there other viable datasets that could be used for a reduced comparison? E.g., rather than all the baselines, just compare MAD-Sherlock with say GPT-4o# and Sniffer on another dataset? Or even comparing with GPT-4o# alone could still significantly confirm the robustness of the results.
> >
> > I'll raise my score a point now, as most of my criticisms have been strongly addressed, and consider another point after reading the other reviews.

---

### Author Response · Authors · 2024-11-28
**Overview of Contributions and Improvements**

We thank the reviewers for their valuable feedback and insights provided on our submission. We are very happy to see a positive reception of our work and to be granted the opportunity to answer any questions that the reviewers might have.

We understand that there might have been some misunderstandings while interpreting our work, therefore, we would like to reiterate our core contributions and describe our experimental setup including the user study.

## Key Contributions:

- We propose using multi-agent debates with external retrieval for the task of out-of-context (OOC) misinformation detection. Our proposed system, MAD-Sherlock, not only accurately detects instances of misinformation but also provides coherent explanations for the same.
- We present a novel LLM-based post training approach for scalable OOC misinformation detection that simultaneously improves contextual reasoning, provides in-built explainability and achieves state-of-the-art detection accuracy even without task-specific finetuning.
- Our system involves the use of an advanced external retrieval module which uses reverse image based search and LLM-based summarization to provide agents with external real-time context related to the input.
- We provide an extensive set of experiments comparing MAD-Sherlock to other related methods and baselines in order to establish the superiority of our method. We show that compared to single-agent chain-of-thought approaches, the use of multiple agents allows for a clean separation of agent contexts, decentralisation of action spaces and the opportunity for parallel computation.
- We provide a comprehensive user study to evaluate the effectiveness of our system in detecting and explaining misinformation and show that insights from our system are able to increase overall accuracy for the task of detecting misinformation.

## New Baseline Analysis:
In response to reviewer feedback, we have added a new baseline to our updated submission: **GPT-4o with external context but without the debate framework**. This baseline isolates the effect of the debate setup, allowing us to directly demonstrate its critical role in MAD-Sherlock’s performance. Results from our updated experiments show that the inclusion of multi-agent debates improves both detection accuracy (from 86% to 90%) and explanation quality. This strengthens our claim that the debate setup is not only beneficial but fundamental to the effectiveness of our method.

We are confident that this addition further substantiates our methodology and its contributions, and we thank the reviewers for inspiring us to conduct this crucial analysis.

---

### Author Response · Authors · 2024-11-28
**Summary of Reviews, Improvements and Clarifications**

We sincerely thank all reviewers for their time, effort, and insightful comments on our paper. Taking all feedback into account, we respond to each review individually.

## Summary of reviews
- Reviewer xyn6 appreciates our exploration of an “important problem” and believe our proposed approach “makes a lot of sense”. They also acknowledge the extensive experiments and ablations we include and believe they can be “potentially informative for methods in other domains too”.
- Reviewer peoS finds our work “very interesting” and acknowledges that our experiment results validate the effectiveness of our proposed method. They describe the paper itself as “written in great detail”.
- Reviewer UiNZ finds our approach to exhibit “significant innovation” that shows “major improvement” over current methods, underscoring the high novelty of our work.
- Reviewer AktV believes that our work offers “valuable insights” for using multi-agent systems for related tasks.

## Improvements and Clarifications
### Improvements
Based on the insightful and valuable feedback received from all the reviewers, we have made specific revisions to the paper. These include:
- Adding a new baseline—**GPT-4o with external context but without the debate framework**—to highlight the importance of our multi-agent debate setup. This comparison shows that the debate framework significantly improves performance, further reinforcing the value of our methodology.
- Further refining and expanding our discussion on the work related to our proposed method
- Deepening the discussion to establish why we pick “debates” for our particular problem
- Including information related to time and cost efficiency
- Expanding our future work section to include efforts to further improve the external information retrieval system

### Clarification on the User Study
Firstly, we would like to clarify that the purpose of the user study is not to convey that insights generated by MAD-Sherlock are better than those from journalists. Rather, our intention with the study is to show that insights from MAD-Sherlock are able to add significant value to the misinformation detection workflow in a fully automated way. This can be important in settings where:
1. Fully automated detection is necessary due to unavailability of human experts
2. Models can assist human experts in making better decisions faster, and can help partially stopgap a lack of human expert availability by uplifting unskilled humans

In our current set-up, we believe internet access could be considered a confounding factor in the study as it introduces variability in participants' ability to search, interpret, and evaluate online content. By restricting internet access, we aimed to isolate the effectiveness of MAD-Sherlock's insights without external influences, ensuring that the results reflect the system's intrinsic utility rather than participants' independent research skills. That being said, in order to allow for a more comprehensive study analysis we have added an additional group to the study which would have internet access. We are currently in the process of conducting the study and would like to include the results in the camera ready version of the paper.

## Conclusion
We are grateful to all the reviewers for having taken the time to carefully understand and review our work. We appreciate the opportunity to refine our work and look forward to further discussion and feedback.

---

### Author Response · Authors · 2024-12-01
**Polite Invitation to Engage with the Rebuttal**

Dear Reviewers,

As the rebuttal period is nearing its conclusion (with the window for comments closing tomorrow, December 2nd), we would like to kindly encourage all reviewers to consider engaging with our rebuttal.

We would particularly like to draw your attention to the new baseline results we have presented, which further strengthen our central claim that multi-agent debate possesses intrinsic advantages. We believe these additional results address some of the key concerns raised during the initial review process and provide valuable insights for evaluating our work.

We greatly appreciate your time and effort in reviewing our submission and look forward to any further feedback you may have.

With best regards,

The Authors

---

### Author Response · Authors · 2024-12-03
**Concluding Remarks**

We sincerely thank the reviewers for their thoughtful feedback, which has greatly helped improve the clarity and rigor of our work. In summary, we have clarified that:

1. The debate setup is fundamental to the effectiveness of our method by adding a new baseline to our updated submission: GPT-4o with external context but without the debate framework. This baseline isolates the effect of the debate setup, allowing us to directly demonstrate its critical role in MAD-Sherlock’s performance. Results from our updated experiments show that the inclusion of multi-agent debates improves both detection accuracy (from 86% to 90%) and explanation quality.

2. The purpose of the user study is not to convey that insights generated by MAD-Sherlock are better than those from journalists. Rather, our intention with the study is to show that insights from MAD-Sherlock are able to add significant value to the misinformation detection workflow in a fully automated way. By restricting internet access, we aimed to isolate the effectiveness of MAD-Sherlock's insights without external influences, ensuring that the results reflect the system's intrinsic utility rather than participants' independent research skills. Therefore, the current setup of the study is completely fair.

3. The external retrieval module can be further refined by using a stronger language model however we do not have evidence to believe that our choice of summarization model had a negative impact on overall performance. We also include information related to time and cost efficiency in order to allow for a more comprehensive evaluation of the system’s applicability to varied use cases.

We note however, that despite our detailed rebuttal, Reviewer UiNZ did not engage with our responses. We believe we have fully addressed their concerns and respectfully request that the other reviewers and the Area Chair consider this when making their final decision.

With best regards,

The Authors

---

### Meta-Review · Area_Chair_7e71 · 2024-12-19

**Metareview:**

**Summary:**

The authors introduce a multiagent multimodal misinformation detection system that focuses on out-of-context (OOC) image usage to produce false narratives. Two or more independent LLM agents debate over several rounds to ideally reach a consensus about whether an image-text pair is misinformation. The results indicate that the system can effectively identify OOC multimodal misinformation and assist human fact-checkers with varying expertise and levels of experience.

**Strengths:**

- This is a simple concept that intuitively should lead to improvement, given prior positive results using multiagent communication to improve LLM reasoning for tasks like mathematical problem-solving. While not particularly technically innovative, this is a relatively novel use case.

-  The LLM explanations of OOC image uses improve transparency of model decision-making and also provide a potential tool for human fact-checkers to identify problematic cases (as concretely shown by Tables 3/4 where their system improves human accuracy).

- The authors seem to have been careful and rigorous in their implementation. Certain details like the debate setup comparison in Table 1 and having agents explicitly point out each other's inconsistencies or ambiguities could inform future multiagent debate research.

- The comparison of laypeople, journalist and academic misinformation detection performance with and without the system in Table 4 is very compelling.

**Weaknesses**

- The system is evaluated on only one (albeit large-scale) dataset.

- Critical ablations and baseline comparisons are missing.

Overall, this is a very interesting paper but not entirely convincing in its present state. It would benefit from polishing and more comprehensive comparisons before publication.

**Additional Comments On Reviewer Discussion:**

The reviews were mixed, and all borderline. The primary concerns appear to be (1) lack of ablations to confirm the effectiveness of the debate and narrow margins in the performance improvement, (2) the inefficiency of multiagent debate, and (3) comparison with a single OOC image-text benchmark (NewsClippings). While this is a well-known and widely used benchmark within the field, it would significantly strengthen the paper's results if the authors can confirm improvement on other evaluation sets. By the author's own admission, their experience with the benchmark "highlights the importance of transitioning to more recent and relevant datasets." For (1), the authors performed a single-agent comparison on a subset of the data, but were not able to perform the full analysis yet. Since this baseline is important to confirm the validity of the paper's findings, I believe these results need to be included and reviewed before publication.

---

### Decision · Program_Chairs · 2025-01-22

Reject